# Infection with IBV DMV/1639 at a Young Age Leads to Increased Incidence of Cystic Oviduct Formation Associated with False Layer Syndrome

**DOI:** 10.3390/v14050852

**Published:** 2022-04-20

**Authors:** Adrea Mueller Slay, Monique Franca, Mark Jackwood, Brian Jordan

**Affiliations:** 1Poultry Diagnostic and Research Center, Department of Population Health, College of Veterinary Medicine, University of Georgia, Athens, GA 30602, USA; adrea.mueller25@uga.edu (A.M.S.); moniquesfranca@gmail.com (M.F.); mjackwoo@uga.edu (M.J.); 2Department of Poultry Science, College of Agricultural and Environmental Sciences, University of Georgia, Athens, GA 30602, USA

**Keywords:** infectious bronchitis virus, DMV/1639/11, false layer syndrome, cystic oviduct, layer

## Abstract

Infectious bronchitis virus (IBV) is an avian coronavirus that causes respiratory disease but can affect the reproductive tract of laying-type chickens. If infection occurs in pullets, false layer syndrome, which is characterized by the development of large, fluid-filled cystic oviducts, can occur. Recently, IBV strain DMV/1639 has been detected in parts of Canada and the U.S., where false layer syndrome has occurred, though it is not clear if IBV is the sole cause or if age at infection is an influencing variable. Our study investigates the role and timing of IBV infection on the development of false layer syndrome, using the IBV types DMV/1639 and Massachusetts (Mass). Six groups of 120 SPF chickens were challenged at either three, seven, or fourteen days of age, using either DMV/1639 or Mass IBV. Cystic oviducts were seen in all the challenged groups, and the pullets challenged at 14 days of age had fewer cystic oviducts than pullets challenged at 3 or 7 days of age. The highest percentage of severe histology lesion scores were seen in the 3-day challenge groups. The data collected in this experiment confirm that IBV DMV/1639 causes cystic oviducts and indicate that age at infection plays a role in the pathogenesis of false layer syndrome.

## 1. Introduction

Infectious bronchitis virus (IBV) is an economically significant pathogen in commercial chickens that causes increased mortality, decreased feed conversion, high condemnation, and decreased egg production and quality [1]. Though mortality rates are low (less than 5%), unless secondary infections are involved, morbidity rates can reach 100% [1,2]. IBV is horizontally transmitted in birds in natural infection [3], and chickens of all ages are susceptible to IBV, though most the severe disease is typically seen in younger birds. Disease severity is dependent upon a combination of factors, including age at exposure, the strain of IBV that the birds are exposed to, and the route of inoculation.

IBV is primarily a respiratory disease; however, some serotypes of the virus are known to affect the epithelial cells of the chicken’s reproductive tract, leading to decreases in egg production and quality. Considerable economic loss can occur in infected hens who pause production and then do not resume normal production levels. IBV also causes poor egg-quality characteristics, such as soft-shelled eggs, mishappen eggs, and eggs with watery albumen [2,4,5]. In addition, false layer syndrome is often described as a clinical sign and the result of exposure to infectious bronchitis virus [3].

The presentation of false layer syndrome, as defined by Sevoian and Levine in 1957, is “a bird with every appearance of a normal bird” that “is out of production and does not lay any eggs”. More specifically, false layer syndrome refers to an absence of egg production of mature layers that are exhibiting little to no other clinical signs [6]. Some birds affected by false layer syndrome exhibit waddling and/or a pendulous abdomen, but many exhibit normal behaviors, including nesting behaviors, vent appearance, and pubic bone characteristics [2,7]. The ovaries of false layers can also appear normal, including empty follicles from recent ovulation. Cyst development in the oviduct with normal ovary development is hypothesized as a cause of these instances of false layers. These cystic oviducts lead to permanent damage that will prevent egg production in the affected birds [8].

Infectious bronchitis virus infection can lead to the development of cystic oviducts, which are strongly associated with false layer syndrome. Cystic oviduct formations with a colorless serous fluid, shorter oviduct lengths, and thin or transparent oviduct walls are common gross lesions of IBV infection [6,7,9,10,11]. Infection with the QX strain (among others) of IBV has been shown to cause fluid-filled cystic oviduct formation, especially in younger birds. These cystic oviducts lead to permanent damage that will prevent egg production in the affected birds. Other IBV strains with a known ability to cause cystic oviducts include the Massachusetts and the Australian T strains [6,12]. Recently, a new strain of IBV, DMV/1639/11, has been associated with cystic oviduct development and false layer syndrome in the United States and Canada, though no causal relationship was shown at the initial time of virus isolation. Therefore, in this work, we describe the ability of the DMV/1639 IBV strain to cause cystic oviducts and false layer syndrome in comparison to a Mass strain known to cause these lesions. We also aimed to determine the effect of the timing of infection on disease severity, both grossly and histologically.

## 2. Materials and Methods

### 2.1. Viruses

The IBV strains used in this study were of the Massachusetts (Mass/41/41) and DMV/1639 (DMV/1639/11) serotypes. The Mass-type IBV challenge is a pathogenic laboratory strain and has been previously described [13]. The DMV-type IBV was isolated from a clinical case of respiratory disease in chickens submitted to the Poultry Diagnostic and Research Center (PDRC) diagnostic laboratory in 2019 and has been previously described [14]. The fourth passage in embryonated chicken eggs was used for the challenge.

### 2.2. Experimental Design

Specific pathogen-free (SPF) chicks were hatched at the PDRC, University of Georgia, and vent-sexed. The female chicks were randomly separated into six groups of 120 birds each. The birds were reared in colony-type houses on fresh pine shavings and given food and water ad libitum. Each colony house was contained and separate from the others. Chicks in select groups were challenged with either the pathogenic Mass (M41) or the DMV/1639/11 IBV strain via the intraocular/intranasal route. Groups 1–3 were given the M41 challenge, and groups 4–6 were given the DMV challenge. Groups 1 and 4 were challenged at 3 days of age; groups 2 and 5 were challenged at 7 days of age; and groups 3 and 6 were challenged at 14 days of age (Table 1).

At seven days post-challenge, choanal swabs were collected from every bird and stored in PBS solution at −80 °C. These swabs were used in quantitative reverse-transcriptase real-time polymerase chain reaction (qRT-PCR) to confirm a successful challenge. Necropsies were conducted seven days post-challenge, five weeks post-challenge, and every following 4 weeks until the end of the experiment at 19 weeks. Ten birds were selected from the groups of interest at the necropsy. Samples of trachea and oviduct were collected and stored in 5% neutral buffered formalin for histopathology submission. Additional samples of trachea and cecal tonsils were taken and stored at −80 °C for additional qRT-PCR if needed. At 19 weeks, all the birds were necropsied. Observations of affected oviducts were noted for each bird in every group. At the final necropsy, five cystic/affected oviducts per challenge group and five unaffected/grossly normal oviducts were chosen for histopathology submission. Tracheas from those ten birds were also submitted for histopathology. The gross and histological lesions for the challenged groups were compared to the established literature for normal oviduct development and histology. The groups were compared to each other to evaluate the challenge timepoint that resulted in the most severe lesions.

### 2.3. RNA Extraction and qRT-PCR

Viral RNA was extracted from 50 µL of swab fluid or tissue homogenate using the MagMax96 total RNA isolation kit (Ambion, Austin, TX, USA) and the KingFisher Automated Nucleic Acid Purification machine (Thermo Electron Corporation, Waltham, MA, USA) according to the manufacturer’s protocols. The AgPath-ID™ One-step RT-PCR kit (Ambion Inc., Austin, TX, USA) was used to perform qRT-PCR assays according to the manufacturer’s instructions. The 25 µL qRT-PCR reaction mixture included 12.5 µL 2× RT-PCR buffer, 10 µM of each primer, 4 µM of each probe, 1 µL of 25× RT-PCR enzyme mix, and 5 µL of viral RNA. The primers and probes corresponding to the Mass- and DMV/1639/11-type IBV were previously published [15]. The qRT-PCR reactions were performed on the Applied Biosystems^®^ 7500 Fast Realtime PCR system (Life Technologies Ltd., Carlsbad, CA, USA) under the following conditions: one cycle of 50 °C for 30 min and 95 °C for 15 min, followed by 40 cycles of 94 °C for 1sec and 60 °C for 60 s (Mass-P), or 40 cycles of 94 °C for 1 s and 53 °C for 60 s (DMV/1639/11-P). Each assay included positive and negative controls for the qRT-PCR reaction and RNA extraction.

### 2.4. Histopathology

The trachea and oviduct tissue collected at the necropsy for histopathological examination were stored in 10% neutral buffered formalin at room temperature for a minimum of 24 h. The tracheal tissue was cut into two pieces for submission: one transverse and the other sagittal. The undeveloped oviduct tissue of birds 7 days post-challenge was transferred directly into cassettes for histological processing. The immature oviduct tissue of the older birds was trimmed as needed to fit into the cassette. The developed oviduct tissue was cut transversely in various sections. The tissue was embedded in paraffin wax and cut into 4 μm slide sections. The slides were stained with hematoxylin and eosin and evaluated with light microscopy.

The slides were scored for lesion severity using the following criteria: inflammation (none, mild, moderate, or severe), presence/absence of exudate in the lumen, and presence/absence of lymphoid nodules. The total scores consist of the sum of each lesion score.

### 2.5. Riboprobe RNAScope Analysis

Twenty samples (including a positive and negative control) were selected for in situ hybridization. The RNAScope^®^ Assay from Wang et al. [16] was performed with either a DMV/1639-specific riboprobe or a Mass-specific riboprobe and followed the Formalin-Fixed Paraffin-Embedded (FFPE) Sample Preparation and Pretreatment procedure, as outlined in the Advanced Cell Diagnostics document 322452.

## 3. Results

### 3.1. qRT-PCR

qRT-PCR was used to measure the viral load per group seven days post-challenge, and is expressed as a CT value, where a lower CT value indicates a higher viral load. The mean CT value for the M41-challenged groups was 26.14, while the mean CT value for the DMV-challenged groups was 24.18. Ninety-nine to 100% of the chicks were successfully challenged in all groups (Table 2).

### 3.2. Gross Lesions

Total cystic and affected oviduct prevalence from all necropsy times is summarized in Table 3. The highest total percentage of cystic oviduct was seen in Group 2 with M41 challenge at 7 days of age with 32% cystic oviduct total. The highest percentage of cystic oviduct for DMV challenge was seen in group 4 challenged at 3 days of age at 24.56% cystic oviduct total. The highest percentages of affected oviducts follow the same trends.

For the M41-challenged groups, cystic oviducts were observed at the 1-month (2/10 birds) and the 3-month (2/10 birds) post-challenge necropsy for the chickens challenged at 7 days of age (Figure 1A). The cysts were most prominent in the caudal end of the oviduct (Figure 1A) and were filled with clear fluid. Cystic oviducts were also observed at the 2-month post-challenge (4/10 birds) and the 3-month post-challenge (1/10 birds) necropsy for the chicks challenged at 3 days of age. Some oviducts were not fluid-filled but were considered abnormal as they were especially thin or appeared to be in the early stages of potential cystic oviduct formation. When these abnormal oviducts also appeared to contain fluid, they were considered cystic (Figure 1B). Some cystic oviducts in these groups were large and the cystic dilatation spanned the entire oviduct (Figure 1C), while other oviducts had milder cystic dilatation isolated to the caudal ends (Figure 1D). Additional lesions seen included free yolk in the coelomic cavity (yolk coelomitis) and two instances of red-tinged fluid in the cystic oviducts.

For the DMV-challenged groups, cystic oviducts were first observed at the 2-month post-challenge (1/10 birds) necropsy in the 7-days-of-age challenge group (Figure 2A). Cystic oviducts were also observed at the 3-month post-challenge necropsy in the 3-days-of-age challenge group (1/10 birds) (Figure 2B) and the 14-days-of-age challenge group (3/10 birds). Affected oviducts without distinct cystic dilatation were present in the 3-month necropsy of both the 7- (1/10 birds) and the 14- (1/10 birds) days-of-age challenge groups. These affected oviducts were thin and translucent but were not dilated and fluid-filled like typical cystic oviducts (Figure 2C).

### 3.3. Histopathology

The most prominent histological lesion observed was inflammatory cell infiltration, primarily lymphocytes, plasma cells, and heterophils, in both the tissue mucosa and the laminar propria. Many oviducts also had edema and aggregates of lymphoid nodules. Exudate consisting of cell debris or eosinophilic material was found in the lumen of oviducts across all the groups and in some instances could be correlated with cystic oviducts grossly (Figure 3 and Figure 4). Additional histology findings in a few oviducts include mucosal atrophy, epithelial hyperplasia, and epithelial sloughing. There does not appear to be a relationship between the age at the challenge and the histology lesion score. The lesion scores were similar across all the groups, with some variation. The groups with the highest percentage of scored oviducts with all lesions were challenged at 3 days of age (Table 4 and Table 5).

### 3.4. RNAScope

Using the IBV riboprobe (both Mass and DMV) and the RNAScope, the virus was detected in multiple tissues. Most notably, the virus was detected in the epithelium of the oviduct tissue in a ten-day-old chick challenged with DMV/1639 at 3 days of age. Of the twenty samples submitted for in situ hybridization analysis, only one oviduct was positive for DMV/1639 in the oviduct epithelium. Positive results were also obtained in the trachea, salivary gland, kidney, ureter, and blood monocytes in the oviduct across the twenty tested samples challenged with either M41 or DMV/1639 (Figure 5).

## 4. Discussion

From this work, we have shown that both DMV/1639 and M41 can cause cystic oviduct development, which is the primary cause of false layer syndrome. The Massachusetts strain M41 has previously been associated with cystic oviduct development and reproductive lesions [12], while the DMV/1639 strain has only recently been associated with the cystic oviduct in both the field and the laboratory challenge [17]. In previous studies, cystic oviducts could be detected as early as 14 days post-challenge, when chicks were infected at one day of age with a QX strain of IBV [11], but more commonly 20–30 days post-inoculation with IBV [18]. Our results support this timeline, where the earliest cysts were seen 28 days post-challenge in the M41 group, challenged at 7 days of age, and 56 days post-challenge in the DMV group, challenged at 7 days of age. It should be noted that because we only selected ten birds per necropsy before the final necropsy, it is possible that cystic oviducts were present earlier in the chicks that were not selected for necropsy. Our results also demonstrate that young birds are particularly vulnerable to the reproductive effects of IBV. The groups that were challenged prior to two weeks of age had higher percentages of grossly affected oviducts than the chicks challenged at two weeks of age. This supports many of the previous investigations into false layer syndrome and cystic oviduct development, where younger birds were affected in greater numbers and experienced more severe gross lesions than birds infected at older ages [2,12,18,19].

Cystic dilatation was most commonly found at the shell gland of the oviducts, though this lesion was present in some magnum and isthmus or continuously throughout the length of the oviduct. The cystic dilatation noted in previous research was often located caudal to the areas of oviduct hypoplasia, such as the level of the suspensory ligament or in the lower portion of the immature oviduct and became larger as the hen got older [8,9,20]. In addition to cystic oviducts filled with clear fluid that had a definitive shape, some fluid-filled but shapeless and undeveloped oviducts were also noted. This also supports previous work demonstrating that infection with IBV causes a range of reproductive lesions and abnormal oviducts, not just cystic oviduct development. These abnormal oviducts can range from underdeveloped, but patent, to a blind sac projecting near the cloaca [20]. Additional oviduct lesions, especially in false layers, include oviducts considerably shorter than normal-length and non-patent oviducts [2]. While most birds experiencing false layer syndrome will have occluded oviducts, some might have normal oviducts that are not functioning [21]. The oviducts of infected birds might be normal in appearance, but the degree of development across an affected flock might not be uniform [9]. In a few cases, IBV-infected birds in this study had free yolk in the coelomic cavity at final necropsy. Because IBV damage affects reproductive tissues, the normal egg-development process is disrupted. Damage to the anterior aspect of the oviduct either prevents the infundibulum from properly enveloping the yolk or causes the infundibulum to eject the yolk via reverse peristalsis [2]. In many cases, authors have described an accumulation of yolk fluid in the coelomic cavity of infected birds. This accumulation indicates that ovulation is occurring directly into the body cavity rather than into the infundibulum [2]. This presentation has also been frequently noted in field cases of false layer syndrome [22].

Historically, microscopic oviduct lesions include significant epithelial cell involvement, as these cells are the primary cells targeted by the virus [18]. These lesions resemble respiratory epithelial cell lesions, including epithelial sloughing, necrosis, congestion, and hemorrhage. Inflammatory cell infiltrates can be present, and there may be edema of the lamina propria or submucosa [7,10]. The muscular layers of the oviduct, the lamina propria, and the mucosal glands are subject to an infiltration of inflammatory cells. Plasma cells, mononuclear cells, and lymphocytes make up most of the infiltrates. The lamina propria might also be affected by fibroplasia and/or edema [3]. The primary lesion seen on slides of infected birds in this study was inflammatory cell infiltration, specifically lymphocytes, plasma cells, and heterophils, which is consistent with IBV infection. Edema was present in many oviducts of all the challenge groups and ages and reflects common post-infection findings after IBV challenge [23].

Lymphoid nodule development varied from a few localized nodules to the severe multifocal presence of nodules. Previous studies demonstrate that lymphoid cell infiltration into the oviduct occurs as early as the third day post-inoculation, which leads to the development of lymphoid nodules in the oviduct wall, especially near blood vessels. The infiltration lesions are less likely to lead to permanent damage than the epithelial cell lesions [18], and the degree of infiltration in the oviducts of birds producing poor-quality eggs is approximately the same as the infiltration in the oviducts of birds producing normal eggs [8]. There did not appear to be a relationship between the presence of inflammatory infiltrates and the age at the challenge, which supports the work of Crinion and Hofstad, who found that age did not play a role in the development of infiltrative lesions in the oviduct [18]. Aside from the presence of inflammation, edema, and dilated lumens, there was little to no indication of the cause of the cyst formation. Benyeda et al. [11] describe few histopathological changes associated with cystic oviducts to explain their formation, which was true for this study as well. In that study, they did not find epithelial damage, excessive secretion, or excessive inflammation in the cystic oviducts of birds infected with the QX strain at 1 day of age.

DMV/1639 was detected in oviduct epithelium using riboprobe and RNAScope analysis. IBV has been detected in the lung and cloacal bursa via in situ hybridization with a riboprobe [24], though to our knowledge no previous study has used in situ hybridization with a riboprobe to locate IBV in oviduct tissue. The detection of IBV in the salivary gland via Riboprobe is further supported by the results of Franca et al. [25], who detected IBV antigen in the salivary gland via immunohistochemistry. Similar to riboprobe analysis, immunohistochemistry has been conducted on oviduct epithelium to target the IBV antigen. The viral antigen localizes to the epithelial cells in the oviduct and is often confined to the cytoplasm, making early diagnosis possible with immunohistochemistry [17,26]. In their 1972 experiment, Crinion and Hofstad showed that viral antigen quantities in the chick oviduct were highest at 6 days post-inoculation. As hen age-at-inoculation increases, the viral antigen becomes more difficult to detect in reproductive tissue. Chickens inoculated at a younger age retain the viral antigen in the oviduct longer than older inoculated chickens [18]. Jones and Jordan [20] hypothesize that the immunological naivety of young birds leads to increased viral persistence in the blood, which in turn leads to increased viral presence in the tissue over a longer period of time than birds inoculated at older ages. The best time to test for viral antigen presence appears to be 5–7 days post-inoculation, based on previous literature [18,20]; however, it is important to note that instances of cystic oviducts without the presence of the IBV viral antigen in the oviduct have been recorded in birds 6 weeks of age [11], and the results of riboprobe analysis may be similar.

## 5. Conclusions

IBV DMV/1639 infection at a young age causes reproductive lesions and cystic oviducts in laying hens, particularly when pullets are infected prior to two weeks of age. IBV DMV/1639 and M41 were detected with the DMV-specific or Mass-specific riboprobe RNAscope technique in the oviduct epithelium, confirming IBV infection as the cause of oviduct damage.

## Figures and Tables

**Figure 1 viruses-14-00852-f001:**
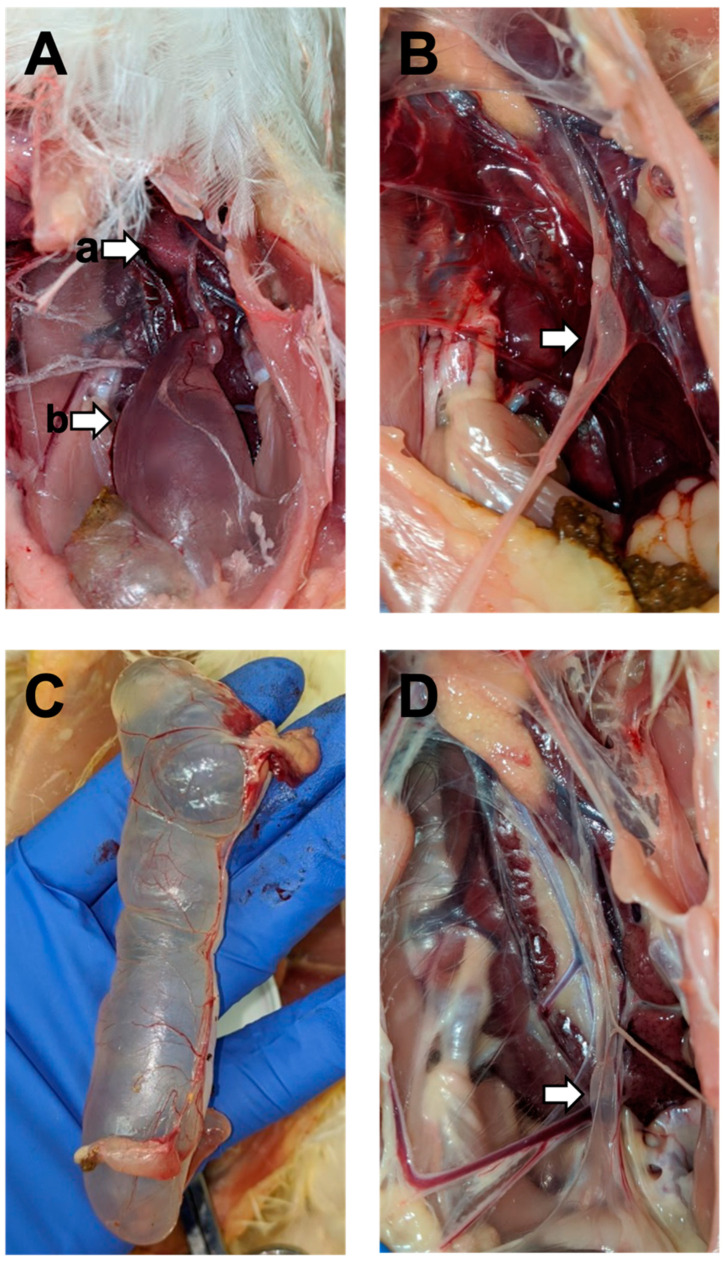
Lesions in the M41 challenge group: ((**A**), (a)) ovary and ((**A**), (b)) cyst in the caudal end of the oviduct from a 5-week-old bird challenged at 7 DOA; (**B**) abnormal fluid-filled oviduct from a 12-week-old bird challenged at 3 DOA; (**C**) large cystic oviduct of a 13-week-old bird challenged at 7 DOA; (**D**) small cyst at the caudal end of the oviduct of a 13-week-old bird challenged at 7 DOA.

**Figure 2 viruses-14-00852-f002:**
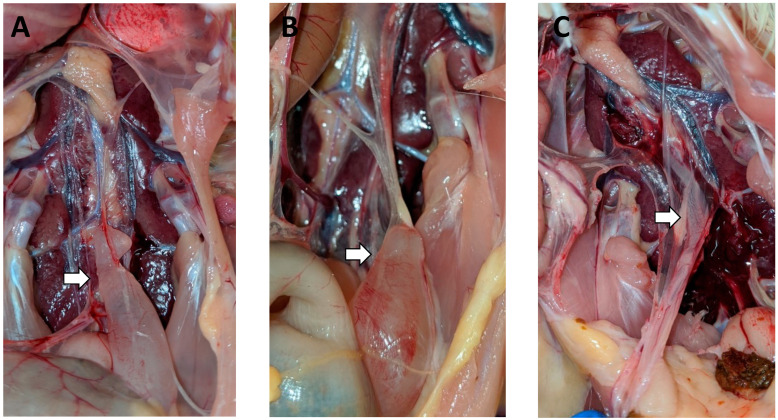
Oviduct lesions in the DMV/1639 challenge group: (**A**) the first cystic oviduct detected in the birds challenged with DMV, seen in a 9-week-old bird challenged at 7 DOA; (**B**) cyst in the caudal end of an oviduct from a 12-week-old bird challenged at 3 DOA; (**C**) small fluid-filled oviduct from a 13-week-old bird challenged at 7 DOA.

**Figure 3 viruses-14-00852-f003:**
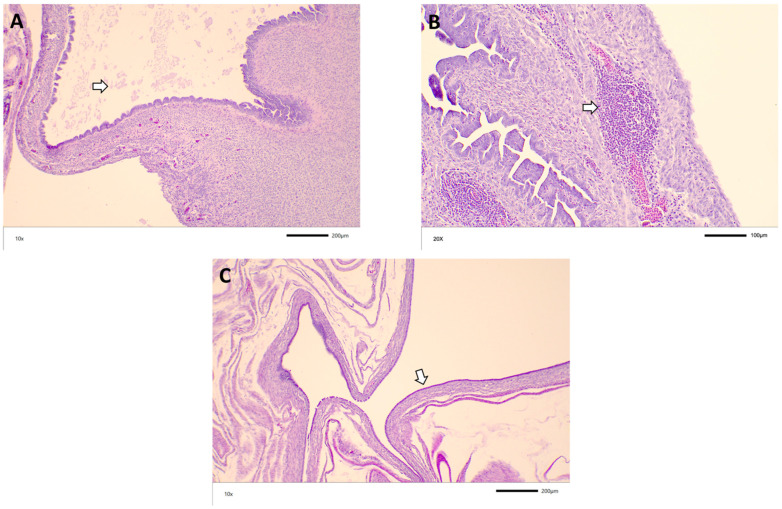
Representative histological findings from M41 challenge: (**A**) exudate in the lumen of the oviduct (arrow) of a 13-week-old bird challenged with M41 at 7 DOA; (**B**) lymphoid nodules (arrow) in the oviduct of a 19-week-old bird challenged with DMV at 3 DOA; (**C**) dilated oviduct with flattened mucosa (arrow) from a 5-week-old bird challenged with M41 at 7 DOA.

**Figure 4 viruses-14-00852-f004:**
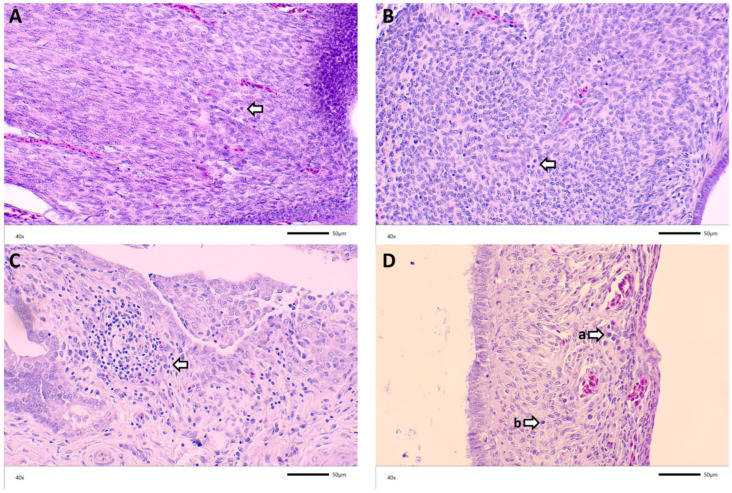
Representative histological findings from DMV/1639/11 challenge: Arrows point to (**A**) minimal inflammatory infiltrates in oviduct of a 2-week-old bird challenged with DMV at 7 DOA; (**B**) lymphocytes in the oviduct of a 5-week-old bird challenged with M41 at 7 DOA; (**C**) lymphoplasmacytic inflammation in the oviduct of a 12-week-old bird challenged with DMV at 3 DOA; ((**D**), (a)) plasma cells and ((**D**), (b)) lymphocytes in the oviduct of a 13-week-old bird challenged with M41 at 7 DOA.

**Figure 5 viruses-14-00852-f005:**
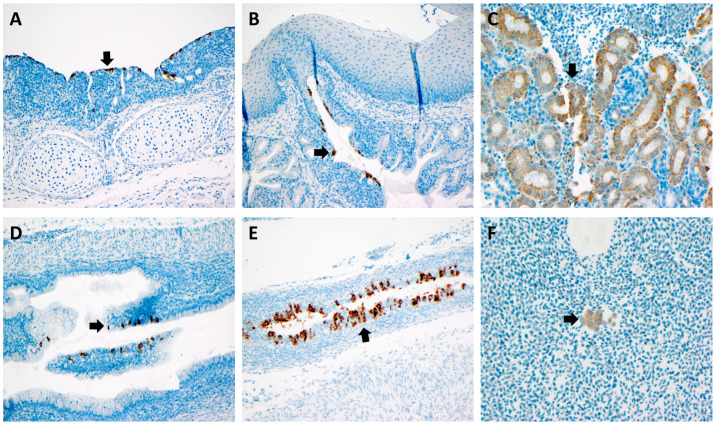
RNAScope findings: (**A**) IBV in the tracheal epithelium (arrow) of a 10-day-old bird challenged with DMV at 3 DOA; (**B**) IBV in the salivary gland epithelium (arrow) of a 10-day-old bird challenged with DMV at 3 DOA; (**C**) IBV in the renal tubular epithelium (arrow) of a 10-day-old bird challenged with DMV at 3 DOA; (**D**) IBV in the ureter epithelium of a 10-day-old bird challenged with DMV at 3 DOA; (**E**) IBV in the oviduct epithelium (arrow) of a 10-day-old bird challenged with DMV at 3 DOA; (**F**) IBV in blood monocytes (arrow) within the oviduct of a 2-week-old bird challenged with M41 at 3 DOA.

**Table 1 viruses-14-00852-t001:** Experimental Design and Summary of Groups.

Group	Challenge	Age at Challenge
1	M41	3 days
2	M41	7 days
3	M41	14 days
4	DMV	3 days
5	DMV	7 days
6	DMV	14 days

**Table 2 viruses-14-00852-t002:** Post-Challenge Viral Load as Measured by qRT-PCR.

Group/Challenge	Age at Challenge	Mean CT Value ± SEM 7 Days Post-Challenge	% Positive Challenged
1/M41	3 days	25.19 ± 2.23	100%
2/M41	7 days	25.85 ± 1.73	99.1%
3/M41	14 days	27.38 ± 2.39	100%
4/DMV	3 days	24.171 ± 2.83	100%
5/DMV	7 days	22.47 ± 2.47	100%
6/DMV	14 Days	25.90 ± 2.08	100%

**Table 3 viruses-14-00852-t003:** Percentage of Cystic and Affected Oviduct per Group.

Group/Challenge	Age at Challenge	Total % Cystic Oviduct	Total % Affected (including Cystic Oviduct)
1/M41	3 days	14.78%	20.87%
2/M41	7 days	32%	34%
3/M41	14 days	9.91%	9.91%
4/DMV	3 days	24.56%	28.07%
5/DMV	7 days	12.28%	13.16%
6/DMV	14 days	8.55%	9.40%

**Table 4 viruses-14-00852-t004:** Histopathology Scores by Lesion.

Score ^1^	Inflammation −	Inflammation +	Exudate −	Exudate +	Nodules −	Nodules +
**Group 1**	43.48%	56.52%	52.17%	47.83%	71.74%	28.26%
**Group 2**	37.5%	62.50%	68.75%	31.25%	64.58%	35.42%
**Group 3**	38.78%	61.22%	53.06%	46.94%	71.43%	28.57%
**Group 4**	51.02%	48.98%	46.94%	53.06%	65.31%	34.69%
**Group 5**	50%	50%	70.45%	29.55%	68.18%	31.82%
**Group 6**	31.91%	68.09%	65.96%	34.04%	48.94%	51.06%

^1^ Histological analysis of slides were broken down into three categories: Presence/Absence (+/−) of Inflammation, Presence/Absence (+/−) of Exudate in the lumen, and Presence/Absence (+/−) of Lymphoid Nodules in oviduct tissue. Birds that were challenged with DMV at 14 days of age had the highest percentage of inflammation and presence of lymphoid nodules. Birds that were challenged with DMV at 3 days of age had the highest percentage of exudate present in the lumen.

**Table 5 viruses-14-00852-t005:** Total Histopathology Scores.

Score ^1^	Total −	Total +	Total ++	Total +++
**Group 1**	23.91%	39.13%	17.39%	19.57%
**Group 2**	16.67%	39.58%	41.67%	2.08%
**Group 3**	16.33%	42.86%	28.57%	12.24%
**Group 4**	12.24%	34.69%	30.61%	22.45%
**Group 5**	34.09%	29.55%	27.27%	9.09%
**Group 6**	14.89%	40.43%	27.66%	17.02%

^1^ Histological scores of slides were combined into total scores. (−) = no lesions seen, (+) one lesion present, (++) two lesions present, (+++) all scored lesions present.

## Data Availability

Not Applicable.

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
