# Peer review of "Infection with IBV DMV/1639 at a Young Age Leads to Increased Incidence of Cystic Oviduct Formation Associated with False Layer Syndrome"

_viruses, 2022, doi:10.3390/v14050852_

Round 1

Reviewer 1 Report

This study was conducted to determine the effect of DMV/1639 IBV strain infection in oviducts of the experimental birds at various times of age. Overall, the paper was written very well and clearly presented. I would ask the authors to address flowing issues that were either not considered or not clearly explained in the paper.

  • The study design does not reflect the study objectives. This study was conducted “to describe the ability of the DMV/1639 65 IBV strain to cause cystic oviducts and false layer syndrome and determine the effect of timing of infection on disease severity both grossly and histologically.” However, the study design included two treatments namely challenge with Massachusetts (Mass/41/41) and DMV/1639 (DMV/1639/11) serotypes and a comparison of pathological changes in the two equal size of treatments groups. Therefore, the given study objective does not necessarily reflect what has been done in the study.

  • Study consisted of a similar size infected controlled group (120 birds x 3 groups infected with Mass/41) to the treatment group (Birds infected with DMV/1639 strain) just for comparison without the inclusion of a non-challenged group (An uninfected controlled group). Mass/41 strain is well known to cause oviduct changes. Therefore, I do not see the purpose of utilizing a large cohort of birds for this controlled treatment. Instead, a portion of birds from this group would have been allocated to include an uninfected controlled group in the study design. The authors should explain the reasons in the paper.

  • Line 82: The colony-type houses were contained from each other?

  • Line 84: Was randomization applied in the selection process of groups for inoculations? The investigator still needs to apply a random method in treatment allocations. For example, name the six groups as A, B, C, D, E, and F. Then select 3 groups randomly for each treatment.

  • Line 92: Was randomization used during bird selection for necropsy at each time point? Or did you select clinically ill birds for necropsy at initial time points?

Typos:

Line 32:  Delete the repeated word “Horizontally”

Author Response

Lines 64-66 have been changed to the following to better reflect the aim of the study: "Therefore, in this work, we describe the ability of the DMV/1639 IBV strain to cause cystic oviducts and false layer syndrome in comparison to a Mass strain known to cause these lesions. We also aimed to determine the effect of timing of infection on disease severity both grossly and histologically."

The reviewer asks about not including a non-challenged group in this study. A non-challenged control group of 50 chicks was kept and placed, but were exposed to challenge virus early in the study. Since this group was no longer non-challenged, we did not use them in the analysis. To overcome this, we compared gross and histological lesions to widely accepted and established literature for normal oviduct development and histology (Lines 100-102).

Line 32: Repeat "horizontally" was removed.

Line 82: "Each colony house was separate and contained from the others" was added to clarify.

Lines 84 and 92: Chicks were sexed after hatch and all female chicks were mixed into a single population. Female chicks were then randomly sorted into 6  baskets with no treatment group assigned to any basket. The baskets of chicks were then placed into the individual colony house in the order of which basket came to hand when going into the colony houses. The word "randomly" was added to line 81 to indicate the random allocation into each group.

Reviewer 2 Report

Viruses 1668379

Infection with IBV CMV/1639 at a young age leads to increased incidence of cystic oviduct formation associated with False Layer Syndrome

Authors: Adrea Muller, Monique Franca, Mark Jackwood, Brian Jordan.

This manuscript summarizes the studies performed to associate IBV infection with two genotypes i.e. (M41; DMV/1639) and the generation of reproductive lesions in the oviducts and false layers. The manuscript is well written and clear and uses direct challenges and gross/histopathology observations to clarify the pathobiology of this condition.

Some suggestions on the pictures and discussion section are given below.

Introduction:

Line 32. There is an extra “Horizontally”

Materials and methods:

Results:

-Images A, B and E are out of focus or are in low resolution

-Nice In situ hybridization images!

Just as a comment, the fact that ISU was used makes me think about other techniques and how they compare. Wouldn’t it be better to do IHC? IHC should be more a bit less sensitive but a lot more accurate since interacts with the proteins rather than a sequence that can be a degraded virus, maybe a good discussion point…

-I suggest having a larger magnification of image F in figure 6. It is not clear if the cell is a macrophage or a lymphocyte.        

Author Response

Line 32: Repeat "horizontally" was removed.

Figure 1 was changed so that panel B was removed as it was out of focus. Panel A and panel E (now panel D in new figure) were kept as the cyst being highlighted by the arrow in the panel is in focus (even if the background is slightly out of focus).

IHC has been previously used to identify IBV spike protein in tissues, and would be an alternative to in-situ hybridization. In this study, we wanted to ensure that our identification of virus in tissues was serotype specific. To make IHC serotype specific, antisera to Mass and DMV/1639 would be necessary. While standardized Mass antisera is available, DMV/1639 is not. To overcome this, we used in-situ as we could customize the hybridization probe to the specific sequence of the IBV. 

The cell indicated by the arrow in panel F of Figure 6 was identified as a monocyte by one of the authors, who is a histopathologist (Dr. Monique Franca). A larger magnification of the image is not available.